# Experimental Study and Calculation Methods of Shear Capacity for High-Strength Reinforced Concrete Full-Scale Deep Beams

**DOI:** 10.3390/ma15176017

**Published:** 2022-08-31

**Authors:** Shushan Li, Ziwen Wu, Junhong Zhang, Wei Xie

**Affiliations:** School of Civil Engineering and Communication, North China University of Water Resources and Electric Power, Zhengzhou 450046, China

**Keywords:** deep beam, high-strength reinforcement, shear strength, shear span-to-depth ratio, strut-and-tie model

## Abstract

The shear behavior of 8 high-strength concrete full-scale deep beams with high-strength steel bars was studied. The depth beam size was 100 mm × 900 mm × 2200 mm, the test parameters included the shear span-to-depth ratio (λ = 0.9, 0.6, 0.3), longitudinal reinforcement ratio ($\rho_{s}=0$.66%, 1.06%, 1.26%) and stirrup reinforcement ratio ($\rho_{\mathrm{sv}}$ = 0, 0.26%, 0.34%, 0.5%). The ratio of the cracking load of the inclined section to the ultimate load is between 30% and 50%, and the bending deformation of the deep beam is small, showing the characteristics of brittle failure for deep beams. Under the action of a concentrated load, the failure mode of deep beams with a small shear span ratio is the failure of the diagonal compression struts, which is very different from that of shallow beams with a large shear span ratio. With the increase of shear span ratio from 0.3 to 0.9, the ultimate shear capacity of deep beams decreases by 19.33%. With the increase of longitudinal reinforcement ratio from 0.67% to 1.27%, the ultimate shear capacity of deep beams increased by 45.02%. With the increase of vertical stirrup reinforcement ratio from 0% to 0.5%, the ultimate shear capacity of deep beams increased by 8.93%. Increasing the area of longitudinal bars or stirrups limited the transverse tensile strain of the compression struts, which is conducive to improving the compressive strength of the compression struts of deep beams and then improving the bearing capacity of deep beams. The strut-and-tie model (STM) is more suitable for analyzing the shear capacity of deep beams. The calculation methods for calculating the shear capacity of deep beams were compared with ACI 318-19, CSA A23 3-19, EN 1992-1-1:2004, Tan–Tan model, Tan–Cheng model, softened STM (SSTM) and simplified SSTM (SSSTM). The results showed that the shear capacity of deep beams could be well predicted by reasonably determining the STM parameters.

## 1. Introduction

Deep beams are widely used as transfer girders, pile caps, pier caps, etc. They are also commonly encountered in bridges and offshore structures [1]. ACI 318–19 defines the deep beams as “the structural members loaded and supported on opposite faces so that compression strut develops between load and support. The maximum length-to-depth ratio of such a deep beam would be approximately 2” [2]. As a result, the increase of shear capacity according to the decrease of shear span–to–depth ratio is more significant in deep beams than in slender beams [3]. Other factors also affect the shear bearing capacity of deep beams, such as concrete compressive strength, beam height, longitudinal reinforcement ratio, stirrup ratio, etc. [4,5,6,7,8,9,10].

The stress form of a deep beam is different from that of a general beam. When diagonal cracking occurs, the shear force of the deep beam is mainly transmitted in the form of arch action [11]. The shear span of the deep beam is classified as a typical D-region (e.g., Figure 1) [12]. There are geometric or static discontinuities in members with D-regions, and the conventional plane section assumption is no longer applicable [13,14]. Chenhui et al. [15] used finite element software to simulate the stress and strain distribution patterns of deep beams and analyzed the force performance of deep beams using a cracked tension-compression bar model. Researchers have conducted a large number of shear tests to investigate the shear behavior of high-strength reinforced concrete deep beams [16,17,18,19]. However, the complex behavior of deep beams observed by experimental research suggests the need to develop specialized design models [20].

The strut-and-tie model (STM) idealizes the discontinuous D-region as a diagonal compression strut from the load point to the support point. It is a more advanced shear design model that simplifies the design and analysis of concrete structures [21]. Varghese and Watstein [22,23] argued that in these models, the bearing capacity of components is regarded as a function of the ratio of the shear span to effective depth (a/d). The shear strength of the deep beam decreases with increasing shear span-depth ratio [24]. The earliest concept of the STM was proposed by Mitchell and Mua [23,25] based on truss analogy field theory. At that time, it was developed on an intuitive basis and was not supported by a corresponding theoretical foundation [26]. As research progressed, the STM based on diagonal compression theory was proposed by Vecchio [27] to predict the shear capacity of members with the D-region.

In recent decades, the STM has been recommended by many national codes to design and analyze the shear strength of deep beams because of its effectiveness. For example, based on the test result of Christ et al., the design theory of bearing capacity design for deep beams in the American ACI 318-83 code was established. Based on the findings of Walther, etc. [28], the bearing capacity design of a deep beam in the European CEB-FIP code was established. Scholars from other countries have also put forward the corresponding design formula and analysis method of the bearing capacity of deep beams through experimental research and theoretical analysis [29]. Three existing codes, namely, CSA A23.3-19 [30], ACI 318-19, and EN 1992-1-1:2004, recommend the STM as the design method for components with discontinuity regions.

Research has continued, and recently many modified or simplified STMs have been proposed. Tan and Cheng established the Tan–Tan model and the Tan–Cheng model, which can simulate the arch mechanism in deep beams and consider the size effect of deep beams [31,32]. Hwang et al. [33] considered the softening effect of a concrete strut under compression loading on the basis of an STM and then proposed the softened STM (SSTM). However, the calculation process included an iterative algorithm that is not convenient in practice. Lu et al. [8] simplified the SSTM by replacing the iterative procedure with a simplified exponential, presenting a simplified SSTM (SSSTM) to render simpler calculations.

Many scholars at home and abroad have carried out a large number of deep beam tests and theoretical studies. However, due to the complexity of its stress and the discreteness of concrete materials, there is no unified theoretical model of shear resistance of deep beam members either at home or abroad. At present, the calculation of shear capacity of concrete deep beam members in China is mainly based on the regression analysis of test results [34]. The simplified strut model is recommended by foreign codes to design the shear bearing capacity of beam members. Therefore, it is of great theoretical significance and practical value to carry out experimental research and theoretical analysis on shear performance of high-strength reinforced concrete deep beam members and put forward a reasonable design and calculation methods. Based on the shear performance test of 8 high-strength reinforced concrete deep beam members, this paper mainly analyzes the failure process and failure mode of such members, and discusses the calculation method of their shear bearing capacity.

## 2. Experimental Study

### 2.1. Materials

Deep beam members were poured with C50 (provided by Pingdingshan Ruiping Cement Co., Pingdingshan, China) commercial concrete. The material test block and deep beam members were poured at the same time. The test blocks are maintained in accordance with the relevant provisions of the Standard for the test method of concrete structures [35] (GB/T 50152-2002) and the Standard for test methods of mechanical properties of ordinary concrete [36] (GB/T50081-2012), and cured at room temperature with each beam for 28 days. Both the test specimens and beams were cured for the same number of days in a controlled environment to make the effect of age on the concrete strength as small as possible. Six 150-mm test cubes were reserved for the compressive test and tensile test. Six 150 mm × 150 mm × 300 mm concrete prism test blocks were constructed to examine the elasticity modulus and the axial compressive strength of the concrete. The mechanical properties of the material were tested by the TYE-3000 press (Wuxi Jianyi Instrument & Machinery Co., Ltd., Wuxi, China). The concrete material test is shown in Figure 2. The test results are shown in Table 1.

In this test, two types of hot-rolled steel reinforcement (provided by Jiangsu Shagang Group, Zhangjiagang, China) were used. HTRB600 grade reinforcement was used as the bottom longitudinal reinforcement, with diameters of 16 mm, 20 mm and 22 mm. HRB400E grade reinforcement was used as horizontal and vertical reinforcement with a diameter of 8 mm. According to the Standard for test methods of concrete structures (GB/T228.1-2012) [37], three diameters of HRB600 bars were selected for tensile testing, and six diameters of HRB400E bars were selected for tensile testing. The parameters measured in the test were yield strain, elastic modulus, tensile strength and yield strength. The tensile test of HRB400E steel was carried out using an electronic universal testing machine CMT5105 (XinSanSi (Shanghai) Enterprise Development Co., Ltd., Shanghai, China) with a range of 100 kN, and the tensile test of HTRB600 steel was carried out using an electrohydraulic universal testing machine WA-500 (Xi’an Minx Testing Equipment Co., Ltd., Xi’an, China) with a range of 500 kN. The material properties obtained from the test are shown in Table 2. The material test process of steel reinforcement is shown in Figure 3.

### 2.2. Specimen Preparation

Eight full-scale deep beams were constructed. The geometry of the specimens was 200 mm wide × 900 mm high × 2200 mm long. Geometrical characteristics and reinforcement details are shown in Table 3. The specimens were analyzed for shear performance in terms of three variables: shear span–depth ratio λ, longitudinal reinforcement ratio $\rho_{s}$ and stirrup reinforcement ratio $\rho_{\mathrm{sv}}$, with the second specimen being a control specimen. The geometric dimension of LDB2 is shown in Figure 4. The effective span-to-height ratio $l_{0}/h$ is 2 in this test.

### 2.3. Test Instrumentation and Procedures

Eight deep beams were loaded to failure by a YJW-10000 pressure testing machine (Shanghai Hualong Testing Instruments Co., Ltd., Shanghai, China) (as shown in Figure 5) with 10,000 kN capacity. The vertical load was transferred to the test piece by means of the distribution beam and the steel packing plate. In the test, the contact area between the deep beam and the upper loaded steel packing plates and the lower support steel packing plates of the deep beam was 80 mm × 200 mm, and the contact area between the steel mat and the deep beam was padded with fine sand to ensure the same force on both sides.

A total of nine strain gauges were placed at different locations on the deep beam surface. Among them, there were three on the diagonal between the mid-span normal and the bearing and loading plates on the right and left sides, which were evenly distributed and used to record the strains in the concrete. This is shown in Figure 6. To accurately determine the stress state of the reinforcement and concrete under static loading, strain gauges were installed at the corresponding positions of the longitudinal reinforcements, horizontal reinforcements and concrete surface. Steel strain gauges were attached in the manner shown in Figure 7. Three displacement gauges were placed both at the left and right supports and at the mid-span position to record the deflection. The details are shown in Figure 8.

The ultimate shear force of the deep beams was estimated with reference to GB 50010-2010 [35]. First, the specimens were preloaded, and all instruments were checked for normal operation. After formal loading, the loading rate was set to 1 kN/s, and each level was loaded with 100 kN, which was roughly graded into 10–20 levels. After each level was completed, the static load was applied for 3 min, and the width of the cracks was measured and recorded to observe the crack extension. When a normal section crack, diagonal section crack and failure were about to appear in the specimen, the loading rate was reduced appropriately until the specimen failure. To facilitate the observation of the final failure pattern of the specimen, the specimen was slowly unloaded to 70% of the peak load after failure.

## 3. Test results and Discussion

### 3.1. Crack and Failure Modes

Table 4 shows the characteristic loads and failure modes of the deep beam specimens at each stage. $V_{\mathrm{cr}}^{N}\;$ is the cracking load of the normal section, which is defined as the load when the first vertical flexible crack occurs. $V_{\mathrm{cr}}^{D}$ is the diagonal cracking load, which is defined as the load when the first diagonal crack occurs. $V_{u}$ is the ultimate load. Table 4 shows that although the parameters of the test members are different, the above three stages were experienced from initial loading to final failure. The cracking load of the high-strength reinforced concrete deep beam section is approximately 20–40% of the ultimate load, while the cracking load of the inclined section is approximately 20–50% of the ultimate load. All specimens of the high-strength reinforcement concrete full-scale deep beam damage modes exhibited diagonal compression failure. Additionally, the reason for LDB1-LDB7 failure was crushing-strut failure, and as LDB8 was not configured with vertical shear reinforcements, the cause of failure was diagonal-splitting.

Figure 9 shows the cracking of all specimens at the mode of failure. The cracks in the shaded area were considered to be the cause of the failure to the beam. In the initial stage of loading, the load was mainly borne by concrete, and the deflection at the mid-span of the specimen was small. Subsequently, as the was load increased, vertical flexural cracks first appeared near the mid-span, increasing in number and extending along the height of the section, but with a small crack width of approximately 0.02 mm. With a continuous increase in load, a diagonal crack appeared in the shear span, which expanded rapidly to form a full-length diagonal crack. The concrete in the compression region was crushed, or the specimen split diagonally along diagonal cracks, and eventually, the loading was ceased.

Taking sample LDB1 as an example, we described the failure process of the high-strength reinforced concrete full-scale deep beams. When the load was increased to 444 kN, a vertical flexural crack first appeared at the mid-span. The crack width was approximately 0.002 mm, and the mid-span deflection was 1.21 mm. As the load was increased, vertical flexural cracks extended upward at the mid-span, while a dozen new flexural cracks gradually formed around the mid-span. The newly created cracks were small in width and spread slowly. When the load was increased to 1000 kN, the partial flexural cracks extended up to 1/9 of the beam height. Partial cracks extended at 5/9 of the beam height. The maximum width of the vertical flexural cracks was 0.2 mm. The tests revealed that the longitudinal tensile reinforcement had not reached yield strength.

When the load reached 550 kN, the first diagonal crack appeared along the line between the loading point and the support point from the left hinged support. The crack length extended to 4/9 of the beam height, and the width was approximately 0.008 mm. At the same time, a diagonal crack also appeared between the loading point and the rolling support on the right side. With increasing load, the diagonal cracks on both sides extended upward, and the width of the crack increased. When the load was increased to 1000 kN, the maximum width of cracks on the right side reached 0.3 mm, and the horizontal reinforcement reached yield. There was a slight sound of concrete splitting when loaded to 1150 kN. To 1169 kN, the concrete in the compression region was crushed along the diagonal at the right side of the sample, and the sample failed. 

Figure 10a represents the variation in the strain in the concrete in the span with the height of the beam during the elastic phase of specimen LDB1. From the figure, the strain of concrete at the mid-span of the specimen did not conform to the plane section assumption, which is consistent with the view of Schlaich et al. [8]. In the elastic phase, the neutral axis gradually moved upward as the cracks continued to develop. The concrete tensile strain in the diagonal section of the shear span shows a “bottle” distribution. Figure 10b depicts the relationship between the concrete strain and concentrated loading on the right side of specimen LDB1. The diagonal cracks first appeared within the shear span. Along with the appearance of diagonal cracks, a redistribution of force occurred within the specimen, forming a force system of tensile-pressure arches, with mainly the concrete between the bearing and loading points being stressed. As a result, a large number of diagonal cracks parallel to the loading and support points appeared in the web region of the beam. Eventually, the concrete compression supports were crushed, and the failure mode was crushed-strut.

Figure 11a shows the relationship between the strain in the third layer of the longitudinal reinforcement of specimen LDB1 and the load. The strains on the reinforcement were small before the concrete cracked, indicating that the concrete was predominantly stressed during the elastic phase. As the load increased, normal section cracks appeared first at mid-span, with a slight increase in strain on the longitudinal reinforcement. At the same time, as the crack expanded at the edge of the support, the stress in the longitudinal reinforcement at the edge increased rapidly and eventually converged with the strain at mid-span, indicating that the longitudinal reinforcement was fully utilized in the high-strength reinforcement concrete full-scale deep beams, forming equal tension tie, and indicating that all longitudinal reinforcement had not reached yield strength. Figure 11b shows the horizontal reinforcement with load. The figure shows that when the specimen failed, all horizontal reinforcements yielded. This indicated that the entire horizontal reinforcement was fully functioning.

Concrete splitting failure occurred at LDB8; as the load was increased, a full-length split crack appeared between the load point and hinged support on the left side of the specimen, and the failure mode was diagonal-splitting. The reason was that LDB8 was not configured with vertical reinforcement. The failure pattern is shown in Figure 12b.

Table 4 shows that the failure mode of all deep beams in this test was the diagonal-compression mode, which shows that this failure mode is common for high-strength reinforced concrete deep beams with small shear span-depth ratios. This mode is characterized by the brittle failure of the concrete under the diagonal cracks and a loud sound at the same time. The longitudinal tensile reinforcement did not reach yield. The horizontal reinforcements all reached the yield strength or were close to the yielding strength. The HRB600 reinforcement was not fully utilized, and the ultimate strength of the specimen was determined by the compressive strength of the preset concrete strut.

There are many factors that affect the shear load capacity of reinforced concrete deep beams, and a large number of experimental studies have shown that the main influencing factors include the concrete strength, section size, shear span-depth ratio, effective span–depth ratio, longitudinal reinforcement, horizontal distribution of reinforcement, hoop reinforcement, loading method, loading speed, etc. [38]. The shear span-depth ratio directly reflects the relationship between the damage of the member along the diagonal crack and the load, and the longitudinal reinforcement and hoop bars improve the shear resistance while also limiting the expansion of diagonal cracks, making the mechanical behavior of deep beams subject to shear more complex. Therefore, three parameters were examined in this test: (1) shear span-depth ratio (LDB1 (λ=0.3), LDB2 (λ=0.6), LDB3 (λ=0.9)); (2) longitudinal reinforcement ratio (LDB4 ($\rho_{s}=0.67\%$), LDB2 ($\rho_{s}=1.05\%$), LDB5 ($\rho_{s}=1.27\%$)); and (3) vertical stirrup ratio (LDB6 ($\rho_{\mathrm{sv}}=0.50\%$), LDB2 ($\rho_{\mathrm{sv}}=0.33\%$), LDB7 ($\rho_{\mathrm{sv}}=0.25\%$), LDB8 ($\rho_{\mathrm{sv}}=0$)).

### 3.2. Load–Deflection Response

According to the test results, the load–deflection response for different λ, $\rho_{s}\;$ and $\rho_{\mathrm{sv}\;}\;$ values are given in Figure 13. From this figure, the mid-span deflection is basically linear until the specimen cracks, and after the diagonal section cracks, it enters the inelastic working phase. Due to the brittle nature of the test and the limitations of the laboratory equipment, the descending section could not be extracted.

Figure 13 shows the relationship between the mid-span deflection and the shear span-depth ratio of the deep beams. The maximum deflections of LDB1, LDB2 and LDB3 were 3.25 mm, 4.66 mm and 6.59 mm, respectively. As  λ  increased from 0.3 to 0.9, the spanwise deflection of the deep beam increased by 102.7%. The failure mode gradually changed from shear compression failure to shear deflection failure, the bending deformation of the member was obvious, and the number of cracks in the shear span area increased. An increase in the number of cracks occurred within the shear-span zone. The stiffness of the specimen was significantly reduced, and therefore, the mid-span deflection was increased. The curve of the rising section of the deflection curve in the load span decreased gradually.

The effect of the longitudinal reinforcement rate on the mid-span deflection is shown in Figure 13b. The maximum deflections of LDB4, LDB2 and LDB5 were 4.44 mm, 4.66 mm, 4.60 mm, respectively. The increase in longitudinal reinforcement improved the flexural properties of the member and increased the stiffness of the specimen so that for the same section height, the increase in longitudinal reinforcement led to a reduction in mid-span deflection. The change in the vertical stirrup ratio had little effect on the flexural stiffness of the specimen, and there was no significant change in the mid-span deflection. The maximum mid-span deflections of LDB6, LDB2, LDB7 and LDB8 are 5.01 mm, 4.66 mm, 5.65 mm and 5.78 mm, respectively.

### 3.3. Cracking and Ultimate Shear Strengths

Figure 14 illustrates the relationship between the characteristic loads and the three parameters. The values of $V_{\mathrm{cr}}^{N}$, $V_{\mathrm{cr}}^{D}$ and $V_{u}$ reflect the shear capacity of the members. $V_{\mathrm{cr}}^{D}/V_{u}$ represents the reserve strength of the deep beam specimens.

The relationship between the characteristic load and λ  is shown in Figure 14a. With the increase of λ  from 0.3 to 0.9, the ultimate bearing capacity of the deep beam decreased by 19.33%. As λ  increased, the diagonal crack extended to an increasing height, which was developed more fully. The inclination angle of the diagonal cracks gradually decreased. This is because the distance between the loading point and the support point increased, and the effectiveness of the tension arch force system decreased, resulting in a decreasing trend in the $V_{\mathrm{cr}}^{N}$, $V_{\mathrm{cr}}^{D}$ and $V_{u}$ of the specimen.

Figure 14b shows the relationship between the characteristic load and longitudinal reinforcement ratio. With the increase of longitudinal reinforcement ratio from 0.67% to 1.27%, the ultimate load capacity of the deep beam increased by 45.02%. The longitudinal reinforcement ratio had a relatively large influence on the ultimate load of the high-strength reinforced concrete deep beam but had little influence on the normal section load and inclined section load. With the increase of longitudinal reinforcement rate, the crack width decreased, the mechanical bite of the aggregate increased, the pinning action was enhanced, and the ultimate load capacity increased.

As the vertical reinforcement rate increased, the height of vertical flexural crack extension and the inclination of the diagonal cracks were basically the same for specimens with the same cross-sectional height. Therefore, with increasing vertical reinforcement ratio, the $V_{\mathrm{cr}}^{N}$, $V_{\mathrm{cr}}^{D}$ and $V_{u}$ values of the specimen were basically unchanged. However, vertical reinforcement can limit the width and the development of diagonal cracks, thereby improving the ductility of deep beams [37].

### 3.4. Reinforcement Strain Response

Figure 15 and Figure 16 show the effect of each of the three variables on the strain of the longitudinal and vertical reinforcement to understand the role these two types of steel reinforcement played in the shear resistance of deep beam members. Before the normal section cracks, their strain was very small, and before the cracking of the inclined section, the shear force was mainly borne by the concrete. With the formation and expansion of cracks in the normal and diagonal sections, the beam produced an obvious redistribution of stresses. From the coexistence of the arch and beam action to the arch action, the longitudinal reinforcement strain increased, forming a “tie arch” stress system with the longitudinal reinforcement as the tie bar and the concrete between the load plate and the support plate as the arch web. As the external load increased, the longitudinal reinforcement basically grew linearly. Under the ultimate load, the longitudinal bars did not reach their yield stress. With the creation of diagonal cracks, the horizontally distributed reinforcement and the concrete together resisted the action of shear force. The strain on the horizontal reinforcement through the diagonal crack increased rapidly and eventually yielded, giving full effect to the horizontal distribution reinforcement.

## 4. Comparisons with STMs

### 4.1. American ACI 318-19 Code

In this code, the calculation formula of the shear strength of a deep beam is as follows:(1)$$V_{n}=F_{\mathrm{ns}}\sin\theta_{s}$$ where $V_{n}$ is the shear bearing force of the deep beam, kN. $\theta_{s}$ is the minimum angle between the concrete diagonal strut and the steel tie connected to it, and the American code stipulates it should not be less than 25°. $F_{\mathrm{ns}}$ is the normal shear force of the concrete strut, which can be calculated by Equation (2):(2)$$F_{\mathrm{ns}}=f_{\mathrm{ce}}A_{\mathrm{ce}}$$ where $f_{\mathrm{ce}}$ is the nominal compressive strength of the concrete strut, MPa, $f_{\mathrm{ce}}=0.85\beta_{c}\beta_{s}f_{c}^{\prime }$.$\beta_{s}$ is the strength reduction coefficient after the cracking of the strut, dimensionless. $\beta_{c}$ is the strength coefficient by the influence of concrete confinement on the effective compressive strength of a strut or node, dimensionless.$\;A_{\mathrm{ce}}$ is the cross-sectional area at the end of the strut under consideration, ${\mathrm{mm}}^{2}$.
(3)$$A_{\mathrm{ce}}=bw_{s}$$ where b is the width of the specimen section, mm. And $\;w_{s}$ is the width of the diagonal strut, mm.
(4)$$w_{s}=w_{t}\cos\theta +l_{b}\sin\theta$$
(5)$$\tan\theta =\frac{h-\left(w_{t}+w_{t}^{\prime }\right)/2}{a}\geq 0.488\;\;$$ where $w_{t}$ is the height of the bottom nodal, mm. $w_{t}^{\prime }$ is the top nodal zone, mm. $w_{t}=w_{t}^{\prime }$.$\;l_{b}$ is the width of the loading steel plate, mm. In addition, the arrangement distributed reinforcement should meet the reinforcements of the minimum requirements ratio, as follows:(6)$$\sum \frac{A_{\mathrm{si}}}{bs_{i}}\sin\theta_{i}\geq 0.0025$$ where $A_{\mathrm{si}}$ is the total cross-sectional area of the i  layer of reinforcement at an angle $\theta_{i}$ with the axis of the compression bar and running through the spacing $s_{i}$ of the compression strut, ${\mathrm{mm}}^{2}$.

### 4.2. Canadian CSAA23.3-19 Code

The calculation process of the STM used in the Canadian code is essentially the same as for ACI 318-19, except for the correction of the effective compressive strength of concrete:(7)$$f_{\mathrm{cu}}=\frac{f_{c}^{\prime }}{0.8+170\varepsilon_{1}}\leq 0.85f_{c}^{\prime }$$
(8)$$\varepsilon_{1}=\varepsilon_{s}+\left(\varepsilon_{s}+0.002\right)\cot^{2}\theta_{s}\;$$ where $\varepsilon_{s}$ is the tensile strain of the longitudinal reinforcement, dimensionless and $\theta_{s}\;$ is the minimum value of the angle between the compression strut and the node.
(9)$$f_{\mathrm{cu}}=\frac{f_{c}^{\prime }}{1.14+0.68\cot^{2}\theta_{s}}\leq 0.85f_{c}^{\prime }$$

When the yield strength of the reinforcement is less than 400 MPa, the calculation is based on Equation (9).

### 4.3. European EC Code

The process of calculating the STM used in the European code is essentially the same as for the American code, with only the values for the vertical distance between the steel tie and the compression strut adjusted as follows. It is also specified that the deep beam should be taken in the range of 45°–75°.
(10)$$\tan\theta =\frac{h-\left(w_{t}+w_{t}^{\prime }\right)/2}{a}$$
(11)$$w_{t}^{\prime }=1.176w_{t}$$

### 4.4. The Tan–Tan Model

According to Figure 17, the calculation formula of the shear strength of the deep beam is as follows:(12)$$\tan\theta_{s}=\frac{h-\frac{l_{a}}{2}-\frac{l_{c}}{2}}{a}$$ where $\theta_{s}$ is the angle between the concrete diagonal strut and the steel tie, $l_{a}$ is the height of the bottom nodal zone, mm and $l_{c}$ is the height of the top nodal zone, mm.
(13)$$V_{\mathrm{dc}}=f_{c}^{\prime }A_{\mathrm{str}}\sin\theta_{s}$$ where $V_{\mathrm{dc}}$ is the ultimate load of concrete when damaged along the diagonal compression strut, kN and $A_{\mathrm{str}}$ is the cross-sectional area of the concrete compression strut,${\;\mathrm{mm}}^{2}$.
(14)$$V_{\mathrm{ds}}=f_{\mathrm{ct}}A_{\mathrm{ct}}\frac{1}{2\cos\theta_{s}}+f_{\mathrm{yw}}A_{w}\frac{d_{w}-0.5l_{c}}{z_{s}}\cdot \frac{\sin\left(\theta_{s}+\theta_{w}\right)}{\cos\theta_{s}}+f_{y}A_{s}\tan\theta_{s}$$ where $V_{\mathrm{ds}}$ is the resistance of the diagonal compression strut to splitting damage, kN; $f_{\mathrm{ct}}$, $f_{\mathrm{yw}}$ and $f_{y}$ are the compressive strength of the concrete, the yield strength of the web and the yield strength of the longitudinal reinforcement, respectively, MPa; $A_{\mathrm{ct}}$ is the cross-sectional area of the concrete compression strut; and $A_{w}$ and $A_{s}$ are the cross-sectional areas of the web and bottom longitudinal reinforcements, respectively, ${\mathrm{mm}}^{2}$.
(15)$$\frac{V_{n}}{V_{\mathrm{ds}}}+\frac{V_{n}}{V_{\mathrm{dc}}}=1$$

### 4.5. The Tan–Cheng Model

On the basis of the Tan–Tan model, the calculation process of the Tan–Cheng model is as follows. The calculation sketch is shown in Figure 18.
(16)$$V=\frac{1}{\frac{\sin2\theta_{s}}{f_{t}A_{c}}+\frac{1}{f_{c}^{\prime }A_{\mathrm{str}}\sin\theta_{s}}}$$ where $A_{\mathrm{str}}$ is the cross-sectional area of the diagonal compression strut, ${\mathrm{mm}}^{2}$, $A_{c}$ is the area of the deep beam cross section, ${\mathrm{mm}}^{2}$, $\theta_{s}$ is the angle between the concrete diagonal strut and the steel tie, and $f_{t}$ is the total tensile stress, MPa, which is expressed as follows:(17)$$f_{t}=\frac{2A_{s}f_{y}\sin\theta_{s}}{\frac{A_{c}}{\sin\theta_{s}}}+\frac{2A_{w}f_{\mathrm{yw}}\sin\left(\theta_{s}+\theta_{w}\right)}{\frac{A_{c}}{\sin\theta_{s}}}\cdot \frac{d_{w}}{h_{0}}+f_{\mathrm{ct}}$$ where $A_{s}$ and $A_{w}$ are the cross-sectional areas of the web and bottom longitudinal reinforcements, respectively, ${\mathrm{mm}}^{2}$; $f_{y}$ and $\;f_{\mathrm{yw}}\;$ are the yield strengths of the longitudinal and vertical reinforcements, respectively, MPa; and $d_{w}$ is the depth from the intersection of the centerline of the web and the diagonal compression strut to the top of the beam, mm.

### 4.6. SSTM

The SSTM proposed by Hwang et al. [39,40] takes into account the softening of the concrete in the compression strut, dividing the force mechanism into a triple superposition of oblique, horizontal and vertical mechanisms, satisfying the equilibrium condition, the deformation compatibility condition and the intrinsic structure relationship. It can be condensed as Equations (18)–(25):(18)$$V_{v}=C_{d}\sin\theta$$
(19)$$V_{h}=C_{d}\cos\theta$$
(20)$$C_{d}=-D+\frac{F_{h}}{\cos\theta }+\frac{F_{v}}{\sin\theta }$$ where $C_{d}$ is the diagonal compression force, kN;$\;V_{v}$ and $V_{h}$ are the vertical shear force and horizontal shear force, respectively, kN; and $F_{v}$ and $F_{h}$ are the tension force in the horizontal ties and vertical ties, kN respectively.
(21)$$-\sigma_{d,\max}=\frac{1}{A_{\mathrm{str}}}\left[-D+\frac{F_{h}}{\cos\theta_{f}}\cos\left(\theta -\theta_{f}\right)+\frac{F_{v}}{\sin\theta_{s}}\cos\left(\theta_{s}-\theta \right)\right]$$ where $-\sigma_{d,max}$ is the maximum compressive stress. It is composed of a diagonal strut, flat strut and strut, as shown in Figure 19.
(22)$$\sigma_{d}=-\zeta f_{c}^{\prime }\left[2\left(\frac{-\varepsilon_{d}}{\zeta \varepsilon_{0}}\right)-{\left(\frac{-\varepsilon_{d}}{\zeta \varepsilon_{0}}\right)}^{2}\right]\;\;\;\;\mathrm{for}\;\frac{-\varepsilon_{d}}{\;\zeta \varepsilon_{0}}\leq 1$$
(23)$$\sigma_{d}=-\zeta f_{c}^{\prime }\left[1-{\left(\frac{-\varepsilon_{d}/\zeta \varepsilon_{0}-1}{2/\zeta -1}\right)}^{2}\right]\;\;\;\mathrm{for}\;\frac{-\varepsilon_{d}}{\zeta \varepsilon_{0}}>1$$
(24)$$\zeta =\frac{5.8}{\sqrt{f_{c}^{\prime }}}\cdot \frac{1}{\sqrt{1+400\varepsilon_{r}}}\leq \frac{0.9}{\sqrt{1+400\varepsilon_{r}}}$$ where $\sigma_{d}$ is the average principal stress of concrete in the d-direction; ζ is the softening coefficient, dimensionless; and $\varepsilon_{d}$ and $\varepsilon_{r}$ are the average principal stresses in the d-direction and r-direction, respectively.
(25)$$V=\left(K_{h}+K_{v}-1\right)\;\zeta f_{c}^{\prime }A_{\mathrm{str}}\sin\theta_{s}$$ where $\varepsilon_{h}$ and $\varepsilon_{v}$ are the averaged normal strains in the h-direction and v-direction, respectively.

### 4.7. SSSTM

Lu et al. [26] simplified the iterative solution of the coefficients of softening based on the oblique, horizontal and vertical mechanisms of the simplified soften strut and tie model. Figure 19 shows the flow of force in the discontinuous areas of the horizontal and vertical tie reinforcement and the resulting idealization of the struts and ties. The process is described in Equations (26)–(33).
(26)$$V=\left(K_{h}+K_{v}-1\right)\;\zeta f_{c}^{\prime }A_{\mathrm{str}}\sin\theta_{s}$$
(27)$$\zeta =\frac{3.35}{\sqrt{f_{c}^{\prime }}}\leq 0.52$$ where $K_{h}$ and $K_{v}\;$ are the horizontal and vertical tie indexes, respectively, dimensionless.

The horizontal tie is as follows:(28)$$K_{h}=1+\left({\bar{K}}_{h}-1\right)\frac{A_{\mathrm{th}}f_{\mathrm{yh}}}{{\bar{F}}_{h}}\leq {\bar{K}}_{h}$$ where (29)$${\bar{K}}_{h}\approx \frac{1}{1-0.2\left(\gamma_{h}+\gamma_{h}^{2}\right)}$$
(30)$${\bar{F}}_{h}=\gamma_{h}\times \left({\bar{K}}_{h}\;\zeta A_{\mathrm{str}}\right)\times \cos\theta$$

The vertical tie index is as follows
(31)$$K_{v}=1+\left({\bar{K}}_{v}-1\right)\frac{A_{\mathrm{tv}}f_{\mathrm{yv}}}{{\bar{F}}_{v}}\leq {\bar{K}}_{v}$$ where (32)$${\bar{K}}_{v}\approx \frac{1}{1-0.2\left(\gamma_{v}+\gamma_{v}^{2}\right)}$$
(33)$${\bar{F}}_{v}=\gamma_{v}\times \left({\bar{K}}_{v}\;\zeta f_{c}^{\prime }A_{str}\right)\times \sin\theta_{s}$$ where ${\bar{K}}_{h}$ is the horizontal tie index with sufficient horizontal reinforcement, ${\bar{K}}_{v}$ is the vertical tie index with sufficient vertical reinforcement, dimensionless, $f_{\mathrm{yh}}$ is the yield strength of the horizontal reinforcement, MPa, $f_{\mathrm{yv}}$ is the yield strength of the vertical reinforcement, MPa, ${\bar{F}}_{h}$ is the balanced amount of the horizontal tie force, kN, and ${\bar{F}}_{v}$ is the balance amount of the vertical reinforcement force, kN.

### 4.8. Comparison of Shear Design Equations

A total of seven different calculation methods based on the STM were used in this study to predict the shear strength of high-strength reinforcement concrete full-scale deep beams, and the results are shown in Table 5. The mean ratios of $V_{u,\exp}/V_{u,\mathrm{cal}}$ for the three code provisions, ACI, CSA and EC2, were 1.331, 1.708 and 1.544, with variances of 0.03, 0.05 and 0.044, respectively. The results show that the shear resistance derived from ACI was close to the experimental results and that there was less dispersion in the data. The results of the EC2 and CSA calculations were relatively conservative, and the dispersion of the data was small. All three code provisions can predict the shear capacity of high-strength reinforcement concrete full-scale deep beams.

For the Tan–Tan model and the Tan–Cheng model, the means of the predicted values were 1.27 and 1.297, and the variances were 0.009 and 0.009, respectively. The Tan–Tan model considers the role of longitudinal reinforcement, horizontally distributed reinforcement and vertical reinforcement in resisting splitting failure in inclined compression bars compared to the original STM, which only considered compression bar concrete under compression. The Tan–Cheng model, in turn, considered the size effect and revised the boundary conditions. Both models predicted values with high accuracy, and the data dispersion was very small; This is the same conclusion as Tan [41]. Thus, these methods can be used to predict the shear capacity of high-strength reinforcement concrete full-scale deep beams.

For the SSTM and the SSSTM, the means of the predictions were 0.90 and 0.99, and the variances were 0.006 and 0.004, respectively. The SSTM is based on the STM and takes into account the effect of tensile stresses in concrete after compression and satisfies both the equilibrium conditions, namely, the deformation coordination conditions and the intrinsic structure relationship for reinforcement and concrete. The SSTM requires an iterative calculation of the concrete softening factor. The SSSTM simplified the iterative process by directly deriving the softening factor with respect to the strength of the concrete, making the calculation process simple. However, for large deep beams, the predictions of these two methods are relatively high, and further research is needed. The conservativeness of SSTM predictions increases with increasing shear span–depth ratio.

## 5. Conclusions

In order to study the damage mechanism of deep beams and the calculation method of shear bearing capacity, this paper systematically conducted an experimental study on eight high-strength reinforced concrete full-scale deep beam members. The Tan–Tang model, the Tan–Cheng model, the SSTM model and the SSSTM model are used to predict the shear bearing capacity of deep beams, and the adaptability of ACI 318-19, EC2 and CSA specifications for calculating the shear bearing capacity of full-scale deep beams is analyzed. The main conclusions are as follows:The failure mode of the deep beam is the crushing of the inclined compression bar. At the beginning, the bending crack appears at the mid-span, and the load is about 16–37% of the ultimate load. Subsequently, inclined cracks appear between the bearing and the loading point, and the load is about 29–52% of the ultimate load. The failure of deep beams is caused by the penetration of oblique cracks, showing the nature of brittle failure.The shear span-depth ratio is the most important parameter affecting the shear capacity of high-strength reinforced concrete deep beams. With the increase of shear span ratio, the shear capacity of deep beams decreases significantly. When the shear span ratio increases from 0.3 to 0.9, the shear bearing capacity decreases by 19.3%. The number of longitudinal bars at the bottom also significantly affects the shear capacity of deep beams. When the ratio of longitudinal bars increases from 0.67% to 1.05%, the shear capacity increases by 14.6%. In addition, the presence of shear reinforcement is critical for controlling crack propagation and providing ductility for deep beams.The effective coefficient is determined by the compressive strength of concrete, but it is also affected by the shear span depth ratio. ACI 318-19, EC2 and CSA codes ignore the influence of these two parameters when estimating the effective concrete strength of the inclined compression bar, thus leading to non-conservative prediction.

In this paper, the shear bearing capacity of full-scale deep beams is calculated according to the formula suggested by Tan–Tan, Tan–Cheng, SSTM and SSSTM can more accurately predict the shear bearing capacity of deep beams. According to the calculation formulas of ACI318-19, EC2 and CSA, the shear bearing capacity of the full-scale deep beam in this paper is calculated. The average value of the test value and the calculated value is about 1.5, which indicates that the calculation formulas of these specifications maintain a balance between reliability and economic rationality. The results will allow an evaluation of the current code provisions and help identify their limitations. It provides a test basis for the revision of national codes.

## Figures and Tables

**Figure 1 materials-15-06017-f001:**
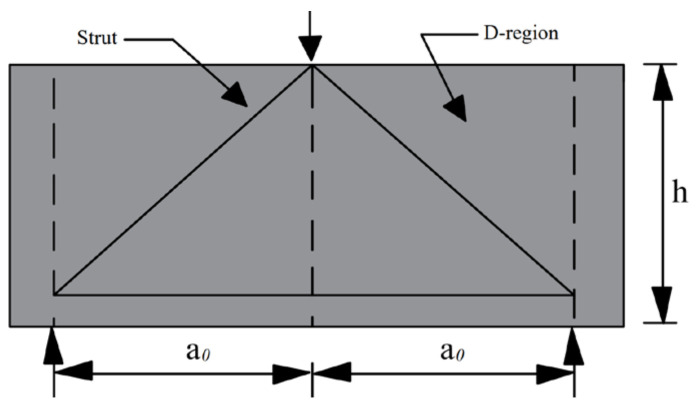
Schematic diagram of deep beam D-region.

**Figure 2 materials-15-06017-f002:**
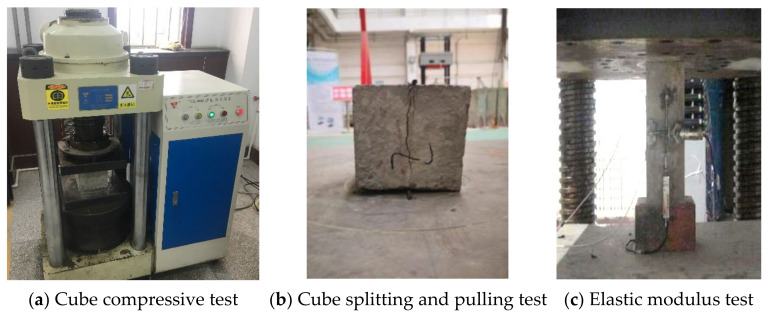
Concrete material test.

**Figure 3 materials-15-06017-f003:**
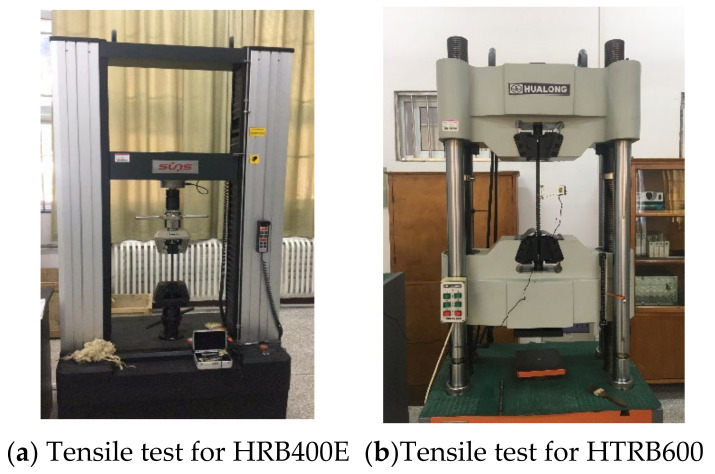
Steel material test.

**Figure 4 materials-15-06017-f004:**
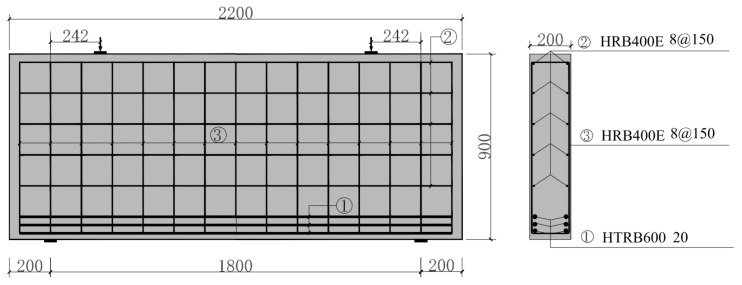
Specimen and reinforcement details (LDB1).

**Figure 5 materials-15-06017-f005:**
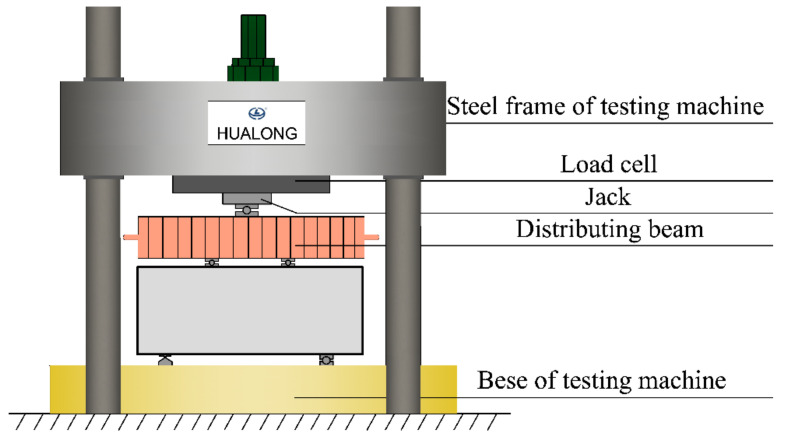
Schematic diagram of loading device.

**Figure 6 materials-15-06017-f006:**
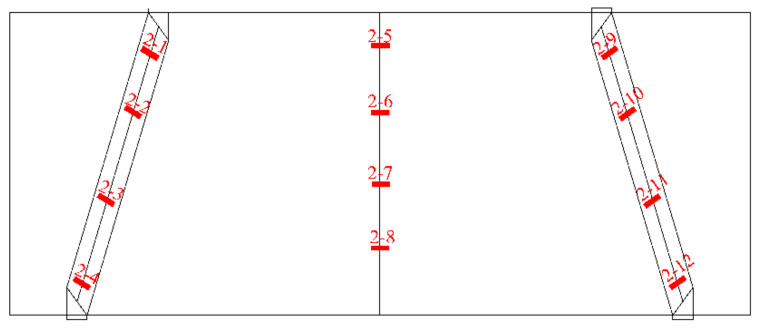
Arrangement of strain gauge of concrete.

**Figure 7 materials-15-06017-f007:**
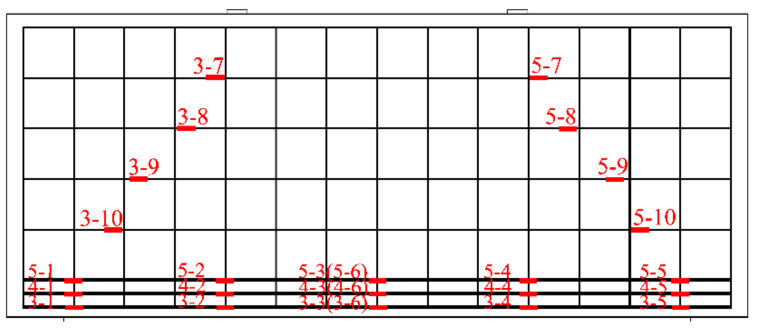
Arrangement of strain gauge of reinforcements.

**Figure 8 materials-15-06017-f008:**
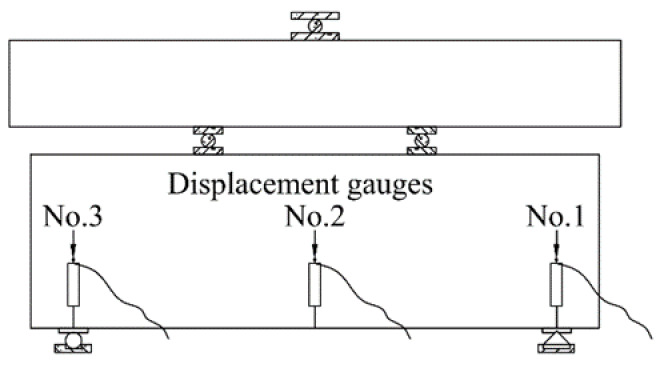
Distribution of meter layout.

**Figure 9 materials-15-06017-f009:**
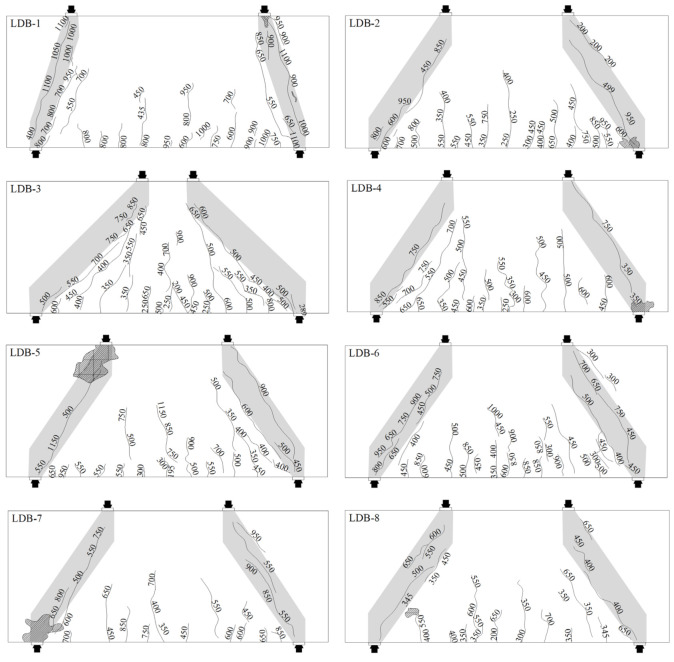
Crack patterns at failure.

**Figure 10 materials-15-06017-f010:**
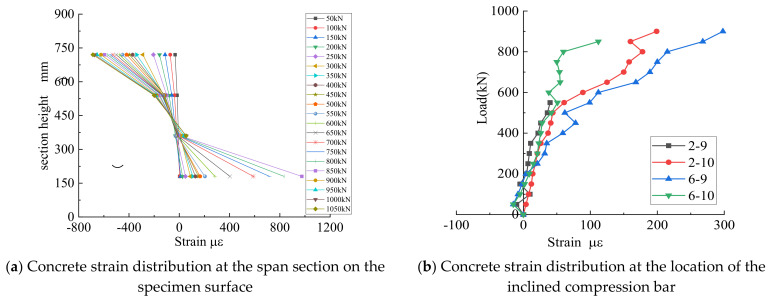
Surface concrete strain of test specimen LDB1.

**Figure 11 materials-15-06017-f011:**
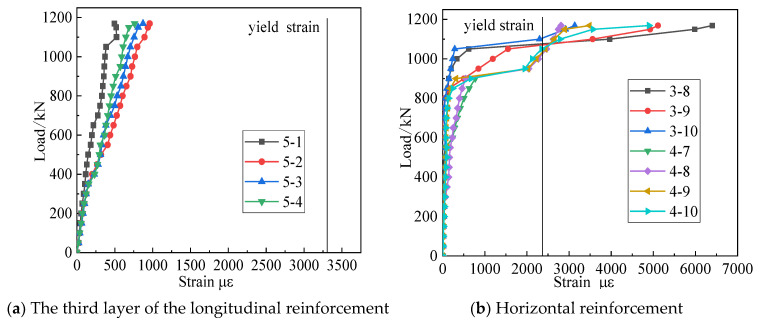
Reinforcement strain of test specimen LDB1.

**Figure 12 materials-15-06017-f012:**
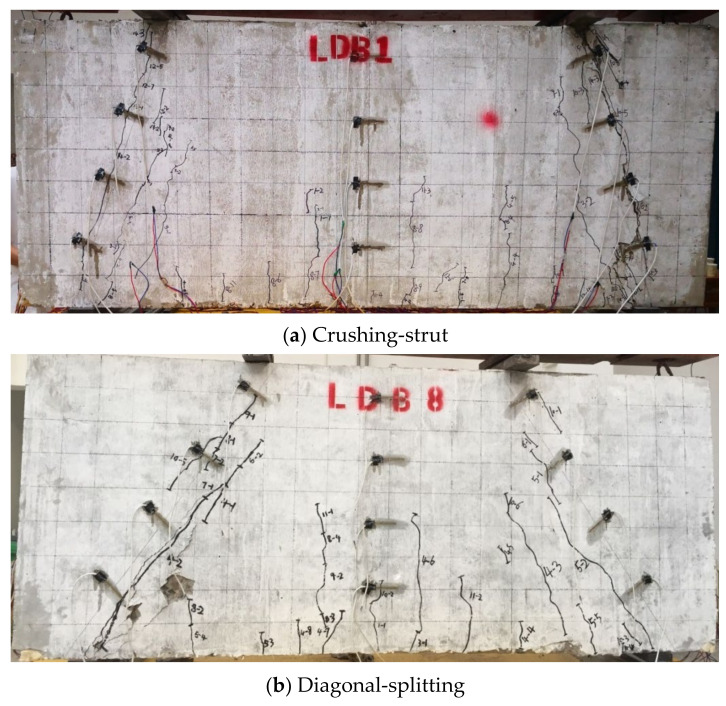
Failure mode of deep beam.

**Figure 13 materials-15-06017-f013:**
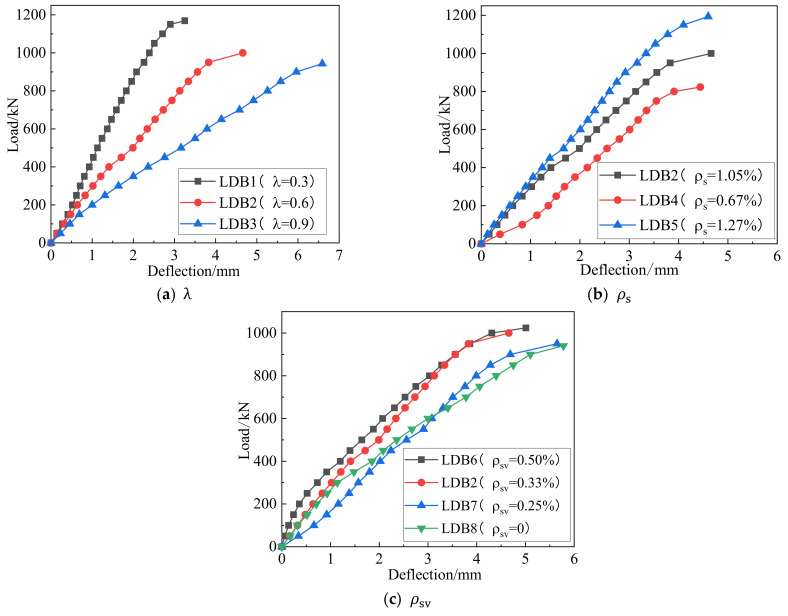
Relation curve between three variables and deflection.

**Figure 14 materials-15-06017-f014:**
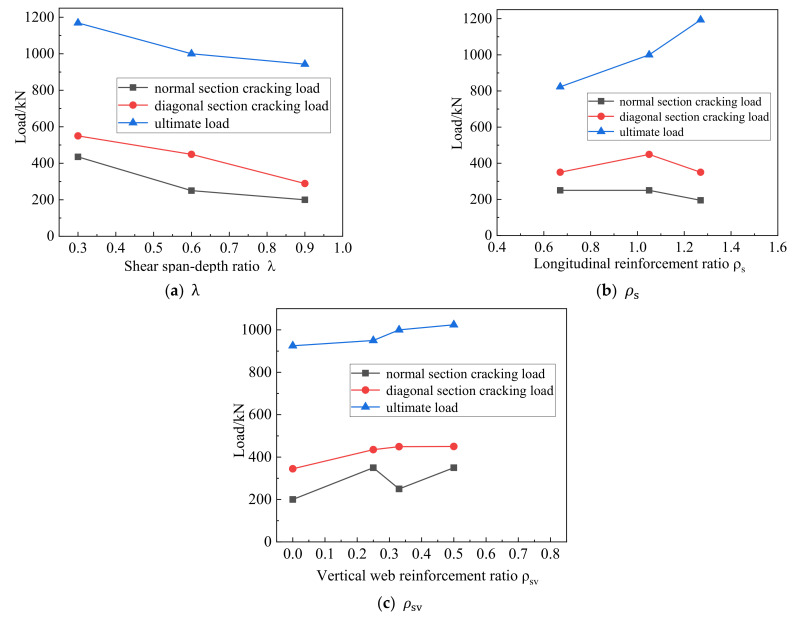
Relation curve between three variables and load.

**Figure 15 materials-15-06017-f015:**
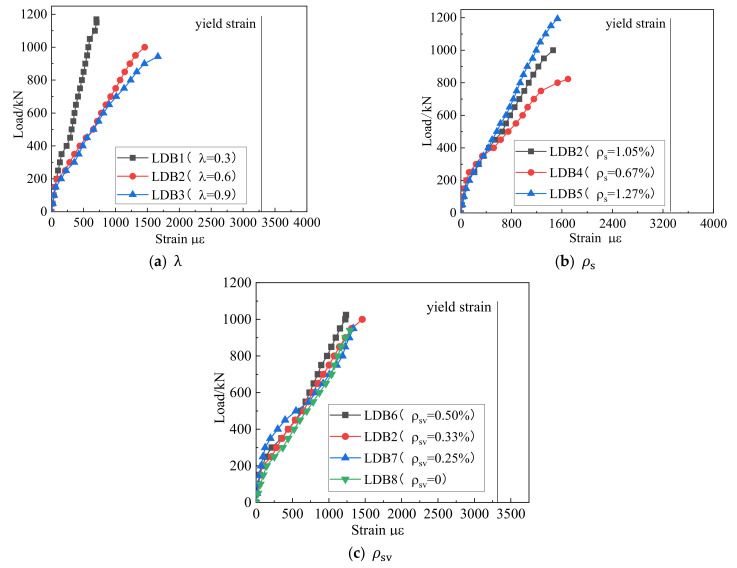
Relation curve between the three variables and longitudinal reinforcement strain.

**Figure 16 materials-15-06017-f016:**
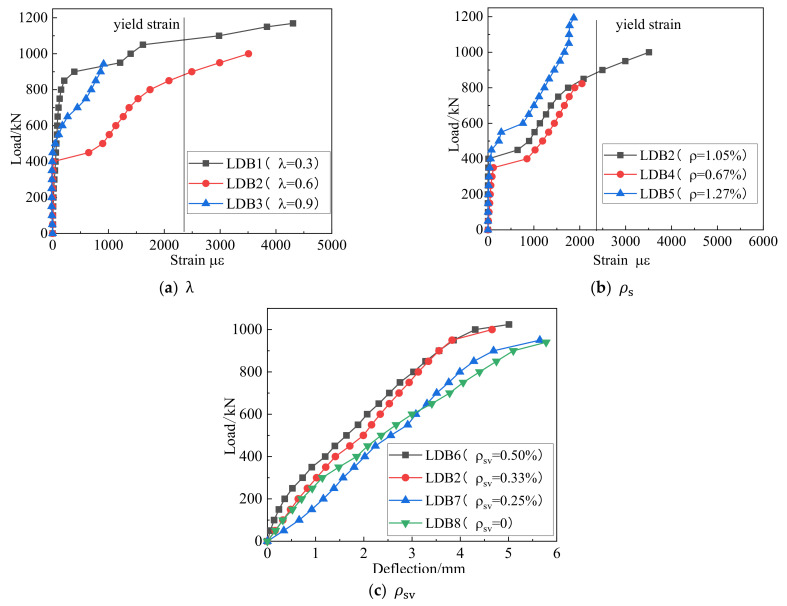
Relation curve between the three variables and horizontally distributed reinforcement strain.

**Figure 17 materials-15-06017-f017:**
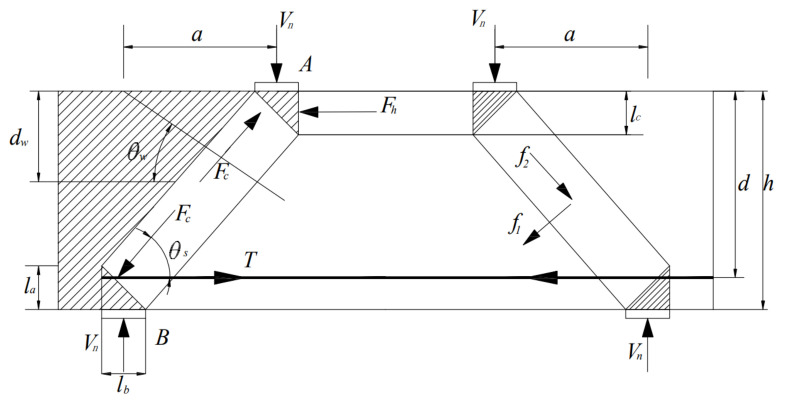
Strut-and-tie model for deep beams.

**Figure 18 materials-15-06017-f018:**
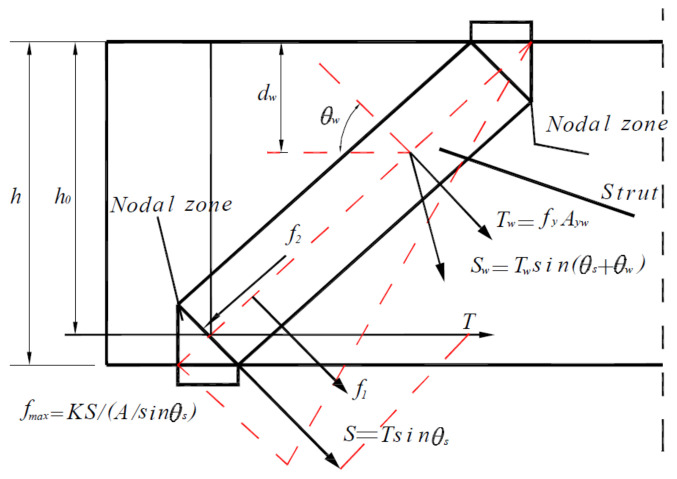
Equilibrium condition of the bottom nodal zone.

**Figure 19 materials-15-06017-f019:**
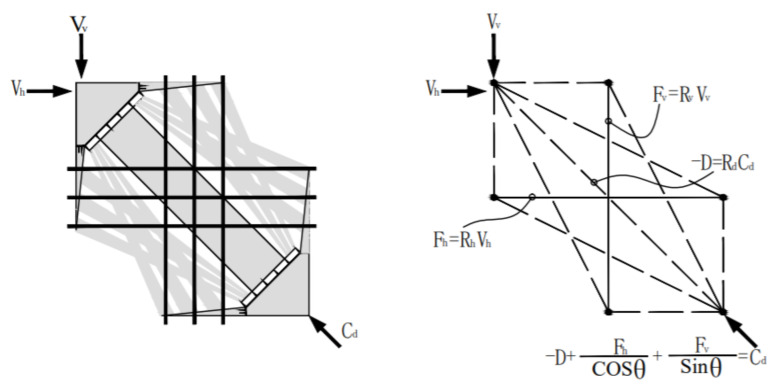
Working mechanism of the SSTM model.

**Table 1 materials-15-06017-t001:** C50 Properties of Concrete Materials.

$f_{\mathrm{cu}}/MPa$	$f_{c}/MPa$	$f_{t}/MPa$	$E_{c}/GPa$
59.82	42.90	3.74	34.64

$f_{\mathrm{cu}}$: cube compressive strength of concrete. $f_{c}\;$: prism compressive strength of concrete.$\;f_{t\;}$: tensile strength of concrete. $E_{c}:$ elastic modulus of concrete.

**Table 2 materials-15-06017-t002:** Steel material properties.

Reinforcement	d (mm)	$f_{y}\;\left(MPa\right)$	$f_{u}\;\left(MPa\right)$	$E_{s}\;\left(GPa\right)$
HTRB600	16	670	865	198.5
HTRB600	20	653.7	823.3	196.6
HTRB600	22	630	800	195.8
HRB400E	8	456.8	647.7	205.3

*f*_y_: specified yield strength for reinforcement. *f*_u_: ultimate strength for reinforcement. *E*_s_: modulus of elasticity of reinforcement.

**Table 3 materials-15-06017-t003:** Specimens design parameters.

Component Number	$l\times b\times h\left(mm\right)$	λ	$l_{0}/h$	$\rho_{s}\;(\%)$	$\rho_{sh}\;(\%)$	$\rho_{sv}\;(\%)$
LDB1	2200×200×900	0.3	2	1.05	0.33	0.33
LDB2	2200×200×900	0.6	2	1.05	0.33	0.33
LDB3	2200×200×900	0.9	2	1.05	0.33	0.33
LDB4	2200×200×900	0.6	2	0.67	0.33	0.33
LDB5	2200×200×900	0.6	2	1.27	0.33	0.33
LDB6	2200×200×900	0.6	2	1.05	0.33	0.50
LDB7	2200×200×900	0.6	2	1.05	0.33	0.25
LDB8	2200×200×900	0.6	2	1.05	0.33	0

*λ* is the ratio of the shear-span ratio as the minimum distance a (a is called shear-span) from the point of concentrated load action on the beam to the edge of the support to the effective height $h_{0}$ of the section.

**Table 4 materials-15-06017-t004:** Test results of LDB series specimens.

Specimen	$V_{cr}^{N}\;\left(kN\right)$	$V_{cr}^{D}\;\left(kN\right)$	$V_{u}\;\left(kN\right)$	$V_{cr}^{N}/V_{u}$	$V_{cr}^{D}/V_{u}$	$\delta \;\left(mm\right)$	Failure Model	Failure Form
LDB1	435	550	1169	37.21%	47.05%	3.25	Diagonal- compression	Crushing- strut
LDB2	250	449	1000	25.00%	44.90%	4.66	Diagonal- compression	Crushing- strut
LDB3	200	289	943	21.21%	30.65%	6.59	Diagonal- compression	Crushing- strut
LDB4	250	350	823	30.38%	42.53%	4.44	Diagonal- compression	Crushing- strut
LDB5	195	350	1193.5	16.34%	29.33%	4.60	Diagonal- compression	Crushing- strut
LDB6	350	400	1024	34.18%	39.06%	5.01	Diagonal- compression	Crushing- strut
LDB7	350	500	950	36.84%	52.63%	5.65	Diagonal- compression	Crushing- strut
LDB8	200	345	940	27.59%	47.59%	5.78	Diagonal- compression	Diagonal- splitting

**Table 5 materials-15-06017-t005:** Comparison of STM predictions and experimental results.

Specimen	Test Value	$V_{u,exp}/V_{u,cal}$						
	Vn (kN)	ACI	EC2	CSA	Tan–Tang	Tan–Cheng	SSTM	SSSTM
LDB1	1169	1.666	2.136	1.563	1.322	1.36	0.89	0.93
LDB2	1000	1.295	1.661	1.489	1.246	1.243	0.92	0.98
LDB3	943	1.303	1.671	1.958	1.377	1.342	1.01	1.09
LDB4	823	1.08	1.385	1.239	1.156	1.112	0.78	0.93
LDB5	1193.5	1.535	1.969	1.767	1.425	1.448	0.99	1.10
LDB6	1024	1.326	1.701	1.525	1.275	1.273	0.96	1.00
LDB7	950	1.23	1.578	1.414	1.183	1.244	0.88	0.93
LDB8	940	1.217	1.561	1.4	1.175	1.35	0.79	0.94
Mean		1.331	1.708	1.544	1.27	1.297	0.90	0.99
Variance		0.03	0.05	0.044	0.009	0.009	0.006	0.004

## Data Availability

Data are contained within the article.

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
