# Peer review of "Experimental Study and Calculation Methods of Shear Capacity for High-Strength Reinforced Concrete Full-Scale Deep Beams"

_materials, 2022, doi:10.3390/ma15176017_

Round 1

Reviewer 1 Report

In this manuscript, experimental tests, theoretical analysis, and model calculations have been used to examine the shear performance of full-size deep beams of high-strength reinforced concrete. Eight full-scale deep beams made of high-strength concrete and high-strength steel bars were tested for their shear behaviour. According to the failure mechanism, the study concentrate on the effects of the shear span-depth ratio, the longitudinal reinforcement rate, and the vertical reinforcement rate on the shear performance. To assess the accuracy of various models in estimating the shear capacity of full-size deep beams of high-strength reinforcement concrete, seven shear calculation models were examined: The Tan-Tan model, Tan-Cheng model, softened STM (SSTM), simplified SSTM (SSSTM), ACI 318-19, CSA A23 3-19, and EN 1992-1-1:2004.

·      In Introduction, related literature is very-well reviewed. In my opinion, it would be helpful to understand if authors also provide some figures/sketches explaining common models, D-region, etc.

·      Is there any numerical studies to investigate behavior such deep beams? If yes, maybe they can be included in literature review.

·      In Introduction, the novelty of the current manuscript is a bit unclear. Could you please highlight the novelty in detail in the last paragraph? Especially, compared to similar studies.

·      I suggest using “Experimental study” for the title of the second section.

·      In Table 1, please provide definitions of abbreviation with the same order given in the table.

·      In Table 1 and the following tables/figures, to indicate SI units, please use parentheses. For example, instead of “d/mm”, use “d (mm)”. Otherwise, one may think it is a ratio of two parameters.

·      Please use “gauge” instead of “gage”.

·      Figure numbering is wrong, there are two “Fig.6”.

·      Figure 7 is too large, it can be put in a single page.

·      In Figure 14, there is a typo: “yeild”

·      In Figure 17, please remove the equation, if it is not necessary.

·      Please check all figure and table numbers and their in-text references.

·      When the computation time and cost be considered, can this study be supported by the computational analysis of mathematical models? In this way, experimental studies and simulation results can be compared too.

·      Why are the results based on SSTM and SSSTM significantly lower than others? Is there any theoretical explanation for this?

·      Regarding my previous comment, it would be ideal to discuss how these codes have been developed. 

Reviewer 2 Report

please, see the attachment.

Round 2
